# Empowerment for the Digital Transformation: Results of a Structured Blended-Learning On-the-Job Training for Practicing Physicians in Germany

**DOI:** 10.3390/ijerph192012991

**Published:** 2022-10-11

**Authors:** Josefin Bosch, Christiane Ludwig, Johannes Fluch-Niebuhr, Dietrich Stoevesandt

**Affiliations:** 1Dorothea Erxleben Learning Center, Medical Faculty of the Martin-Luther-University Halle-Wittenberg, 06112 Halle (Saale), Germany; 2Department for Internal Medicine, University Medicine Halle, 06120 Halle (Saale), Germany; 3Carl-von-Basedow-Klinikum Saalekreis gGmbH, 06217 Merseburg, Germany

**Keywords:** digital competence, digital transformation, continuous training, empowerment, physicians

## Abstract

(1) Background: Practicing physicians have not been in the focus of structured qualifications in basic digital competences so far. However, they are the current gatekeepers to implement digital technologies and need empowerment to proactively take part in the ongoing digital transformation process. The present study investigates if a structured blended-learning training for practicing physicians in Germany enhances both physicians’ knowledge about central aspects of the digital transformation (including awareness of personal possibilities to act) and their attitudes towards a more digitally empowered mindset. (2) Methods: Participants (*n* = 32) self-assessed their knowledge (19 items, 10-point Likert-scale) and attitudes (6 items, 5-point Likert-scale) towards the digital transformation at the beginning and at the end of the training. MANCOVAs were conducted. (3) Results: Participants reported an increase in every knowledge domain, representing large effects (Hedges’ *g* 1.06 to 2.82). Attitudes were partly shifted towards a more empowered mindset with decreased insecurity towards technological, legal, and ethical aspects of the digital transformation (Hedges’ *g* −0.82 to −1.40). However, preparedness for the digital transformation remained low. (4) Conclusions: Generally, the hypotheses were confirmed. The presented on-the-job training had the desired effects on practicing physicians’ knowledge and attitudes. Nevertheless, additional empowerment and support are essential.

## 1. Introduction

The digital transformation in medicine is a change process that not only reshapes the technologies that are used but also the way health care professionals work and interact with patients [1,2,3]. Applying new and digital tools is not a singular technical process. At the same time, it changes the culture and the way of interacting with other health care professionals and patients as well as their relationships with each other [1,2,3,4]. One expectation towards the digital transformation is that patients are empowered to actively take part in their patient journey and to manage their own health status and processes [3,5,6]. We argue that, at the same time, “digital empowerment” is also necessary for health care professionals, with a special focus on physicians in this paper [2,3,7,8,9]. Physicians play a central role in patient care. They have to advise patients and provide them with modern medical care [10,11,12,13]. However, at present, physicians do not feel well-prepared for the challenges of the digital transformation and express a high demand for structured continuing training [13,14,15,16,17,18,19,20,21].

The need for structured qualifications in digital transformation competencies has internationally been acknowledged in the medical education domain [1,2,3,4,9,14,22,23,24]. It is agreed upon that certain digital competencies are necessary for guiding health care professionals and patients through the change process orchestrated through the digital transformation. The term digital competence is inconsistently defined and is highly related to digital literacy [25,26]. As a possible definition, competence is understood as the integration of knowledge, skills, and attitudes [27,28]. Digital competence is differently characterized in a variety of frameworks and definitions [18,25,26,28,29,30]. The European Commission refers to digital competence as one of eight key competences for lifelong learning and describes it as “*the confident and critical use of information society technology (IST) for work, leisure and communication*” [8,28]. Internationally, many programs and efforts invest in the implementation of digital competencies and data literacy into the undergraduate medical curricula [9,24,31,32]. In addition, specialized graduate studies in subjects related to digital health care are offered [9].

The current transformation process requires empowered physicians to now take an active role in the ongoing change process and not to be overrun by it. However, findings and programs for practicing clinicians and their continuing training in basic knowledge about digital transformation processes remain sparse. In Germany, there are few examples, e.g., [33], which implements a basic curriculum proposed by the German Medical Association [34]. For those medical professionals who are currently practicing, lifelong learning has been stated as a goal for many years. However, a “learning by doing” [35] approach with the hope/expectation/demand/request/plea in the direction of doctors to self-qualify in digitalization will not lead to a highly qualified doctoral workforce. Instead, we should focus on practicing doctors as well as on every other group in the health care system and offer them structured guidance. Empowerment for doctors is highly recommended and inevitable, as they are the gatekeepers for the use of digital technologies both for patients and for medical students as well as for informed and critical decisions about therapies, diagnostics, technologies, software, and processes [10,11,12,13,28,36].

According to the Oxford Learner’s Dictionary, empowerment is “*the act of giving somebody more control over their own life or the situation they are in*” [37]. However, what does it mean in regard to physicians? One definition of empowered physicians was offered by Mesko and Győrffy [7]. We would like to highlight three important aspects of their definition. First, physicians play a vital role in empowering patients. Second, they need to be able to use digital technologies, and third, they need knowledge about digital health technologies and legal regulations and have to reflect their patients’ needs.

In the present study, empowerment for the digital transformation is defined as including all actions that support physicians’ control over their professional situation in a digital health care environment. Therefore, physicians need to have competencies in the technological and cultural changes related to the digital transformation in order to make them happen in a human, effective, and healthy way [24,28,30,38,39]. Additionally, physicians both need to be aware about their possibilities to act, design, and shape the current and future digital health care environment and feel confident to move in their patients’ and their own best interests within this environment [2,3,8,13,22,24,25,28]. A better knowledge base, an ability to participate proactively in a change process, as well as a preparedness of physicians is associated with several positive outcomes for both physicians themselves and their patients [25,35,40,41]. For example, a higher knowledge base of physicians about digital technologies is associated with improved patient outcomes [35], higher quality of data resources [42], reduced stress in physicians [7,43,44], and higher workplace satisfaction [45]. High preparedness improves the adjustment to new situations and reduces procedure time, medical errors, and hospital costs [41,45,46,47,48]. That is why in this paper, we would like to highlight the importance of continuing education for practicing physicians to empower them for the ongoing digital transformation [3,22]. We need to raise consciousness in medical education, politics, and industries that physicians are a relevant target group for training in digital competencies [2,3]. We are currently not aware of studies that investigate continuing education for practicing medical professionals with digital transformation as the educational objective itself and how they acquire basic digital competence. That is why in the current study, we present the results of a structured blended-learning on-the-job training for practicing physicians in Saxony-Anhalt, Germany.

In the current study, we focused on two self-evaluated indicators of physicians’ competence for the digital transformation processes: first, practicing physicians’ knowledge about the topics of digital transformation, particularly the knowledge about personal possibilities to act as a reference to empowerment, and second, we analyzed practicing physicians’ attitudes related to their personal empowerment for the digital transformation. We did not include skills as part of the concept of competence [27], as they are either highly common (e.g., computer literacy, navigation literacy as part of digital health literacy [49]) or highly context-specific (related to the medical discipline and the technical equipment of the work place). We formulated the following hypothesis:

**H1.** 
*The structured blended-learning on-the-job training enhances the self-evaluated knowledge base of practicing physicians regarding the central aspects of the digital transformation, particularly the knowledge about their possibilities to act in a digitalized health care environment.*


**H2.** 
*The structured blended-learning on-the-job training changes practicing physicians’ self-evaluated attitudes’ regarding the digital transformation towards a more empowered mindset (i.e., reduced feelings of insecurity, enhanced feeling of preparedness, enhanced reflection on opportunities and risks, and enhanced openness for innovative technologies).*


We carried out a cross-sectional analysis to test our hypothesis.

## 2. Materials and Methods

**Participants and setting.** Participants were physicians from the university hospital Halle (Saale) as well as surrounding hospitals who took part in the blended-learning training “Digitalization in Medicine” at the Dorothea Erxleben Learning Center (DELH) of the Martin Luther University Halle-Wittenberg. The blended-learning training was open to all medical specializations and doctoral positions and took place three times during August 2021 and March 2022 (T1: August 2021, T2: November 2021, T3: March 2022). The Dorothea Erxleben Learning Center—among others—is responsible for the structured continuing education of physicians at the university hospital in Halle (Saale) and is a central stakeholder in the education of health care professionals in the federal state of Saxony-Anhalt. The training was developed and conducted as part of a funding program by the Stifterverband and Daimler Fonds.

**Description of the training.** The curriculum of the blended-learning training was developed in an outcome-oriented way in accordance with the Kern’s Six-Step Approach to Curriculum Development [50]. All steps of the process were consequently oriented to the needs of practicing physicians. Four physicians were involved in the whole process of development and implementation as well as a psychologist.

At the end of the training, participants should have a basic understanding of central concepts and developments of the digital transformation and their interrelation as well as their impact on their own work. They should be enabled to reflect critically on chances and risks of the digitalization during their daily medical work and to advise their patients solidly in favor of their needs. Finally, the training aims to raise awareness to develop a personal position towards the digital transformation and to actively take part in this process. According to the determined needs of the target group, the blended-learning training was based on the inverted classroom model [51] and consisted of three parts (see Figure 1). First, it started with a synchronous online kick-off workshop as an introduction to the topic and the educational objectives of the training. In addition, participants were invited to share experiences and expectations to activate prior knowledge and elicit individual learning goals. Second, participants passed a three-week asynchronous online learning phase, during which they acquired theoretical and background knowledge about central topics of the digitalization in medicine in a self-directed way. They flexibly completed the following six modules on a modern multimedia online platform: (1) important principles and definitions; (2) telematics infrastructure; (3) digital tools; (4) artificial intelligence and big data; (5) ethics in digitalization; and (6) physicians and patients. The modules contained structured micro learning units [52] conveying practical knowledge relevant in the daily work environment of the physicians, mainly by means of video interviews with local experts and practitioners. Complementarily, the online platform contained suggestions to individually deepen the knowledge on certain topics. In addition, a knowledge quiz was offered as a formative self-assessment of the acquired knowledge. The online learning phase was passed parallel to the participants’ job in hospital.

Third, at the end of the training, there was an in-person one-day workshop with a group of maximum fifteen participants. Building on the knowledge base from the online learning phase, participants were encouraged to critically reflect on the topics and to apply their knowledge in simulated patient situations as well as in group discussions. Finally, expectations and learning aims from the kick-off workshop were matched at the end of the course. All training contents were regularly monitored and updated according to the current political and technological development (agile curriculum). The training was certified by the federal medical association of Saxony-Anhalt, offering CME credits. The ethics committee of the faculty provided a certificate of compliance to the local data protection guidelines.

Besides the general and targeted needs assessment, the presented curriculum is based on the above-mentioned basic curriculum by the German Medical Association [34]. Mainly, the topics of both curricula are comparable, and both are certified for 24 CME points. Whereas the German Medical Association integrates its topics into a basic and an advanced module, starting with the telematics infrastructure, the content of the presented curriculum is grouped according to six modules. The first module conveys basic IT-related principles (e.g., interoperability, data security, databases) as premises for comprehension of the other topics. The curriculum in Saxony-Anhalt additionally considers local expertise, such as clinical decision support systems and use cases developed as part of the Medical Informatics Initiative, which is funded by the German Federal Ministry of Education and Research.

**Measures.** In three cross-sectional studies, participants answered the same non-validated questionnaire both at the beginning and at the end of the blended-learning training. Nineteen items regarding the knowledge about content domains of the digital transformation were assessed on a ten-point Likert scale by the participants themselves (1 = very low knowledge, 10 = very high knowledge). These domains were identified as relevant for the target group during the general and the targeted needs assessment [50] and included the knowledge about personal possibilities to act in the digital transformation. Starting with the second training, eight items regarding the physicians’ attitudes were added to the questionnaire, measured on a five-point Likert scale (1 = I do not agree, 5 = I fully agree). Six of them were included in this study and were related to the personal feeling of empowerment for the digital transformation. Additionally, participants were asked to indicate their gender (female/male/divers/not specified) as well as their professional position (resident physician/specialist/senior physician/chief physician).

In the pre-training condition, the questionnaire was conducted online at the end of the kick-off workshop via Evasys (Electric Paper Evaluation Systems GmbH, Lüneburg, Germany). A paper–pencil version was distributed at the end of the one-day workshop for the post-training condition. In the March 2022 course, an online version of the post-test was offered additionally. Participation in the survey was voluntary.

**Data analysis.** We calculated descriptive statistics and checked for missing values as well as normal distribution of the dependent variables. Missing values were minimal (<5% of each variable) and may be interpreted as neglectable [53]. Dependent variables were normally distributed within the range of +/−2 for kurtosis and skewness [54]. Hedges’s *g* was calculated as standardized mean difference per item, indicating effect sizes [55], using the online software of Lenhard and Lenhard [56].

To test each of the two hypotheses, we conducted a multivariate ANCOVA with condition (pre-training vs. post-training) as an independent variable and gender as covariable. Position was not included as covariable owing to small group sizes in the present sample. In the first MANCOVA, the nineteen knowledge items served as dependent variables. In the second MANCOVA, the six items regarding attitudes were included as dependent variables. Eta-squared was calculated as effect size in the MANCOVA.

To ensure anonymous participation in the questionnaire, it was not possible to link the participants data of the two conditions (pre-training vs. post-training). That is why the two data sets were treated as independent samples. All analyses, if not indicated otherwise, were conducted using IBM SPSS Statistics for Windows, Version 25.0.

## 3. Results

**Sample.** In sum, *n* = 32 physicians completed the training. Table 1 shows the distribution of participants during the three training sessions also regarding gender and professional position. Owing to the pandemic, not all starting physicians completed the training. Therefore, the frequencies in the pre-training condition may slightly differ from the numbers indicated in Table 1.

**Descriptives.** The descriptive values and standardized mean differences for self-evaluated knowledge and attitudes in the pre- and post-training condition are presented in Table 2. Figure 2 and Figure 3 additionally illustrate the differences between the two conditions for knowledge (Figure 2) and attitudes (Figure 3) as a function of gender.

**Hypotheses H1—Knowledge.** There was an overall multivariate effect for condition in the MANCOVA, pointing out differences in self-evaluated knowledge before and after the training (Wilks’ lambda = 0.190; *F*(19, 37) = 8.29; *p* < 0.0001). Gender as a covariable did not reach significance (Wilks’ lambda = 0.689; *F*(19, 37) = 0.88; *p* = 0.606).

The between-subjects analysis showed the results presented in Table 3. In every domain, the knowledge was enhanced after the training, including the awareness of participants’ personal possibilities to act (*p* < 0.0001). Depending on the content domain, physicians reported low or medium levels of knowledge before training (minimum: legal aspects of the telematics infrastructure with *M* = 2.4, *SD* = 1.2; maximum: opportunities of digitalization with *M* = 4.7, *SD* = 2.1). After training, the perceived state of knowledge was substantially enhanced and ranged between *M* = 6.2; *SD* = 2.1 (potential applications of artificial intelligence) and *M* = 7.7; *SD* = 1.7 (basics of the telematics infrastructure) and *M* = 7.7; *SD* = 1.5 (telemedicine and online consultation), respectively. Standardized mean differences were *g* > 1.00 for all content domains, and for seven variables, it exceeded *g* = 2.00 (see Table 2), suggesting large effects, according to Cohen [57]. The effect sizes of partial eta-squared *η_p_^2^* were in line with that. Depending on the knowledge domain, condition explained a proportion of 26% to 69% of the variance in the data (see Table 3). For the knowledge about personal possibilities to act, the model explained a proportion of 46% of the variance.

**Hypothesis H2—Attitudes.** The MANCOVA showed an overall multivariate effect for the independent variable, indicating differences between pre- and post-training (Wilks’ lambda = 0.575; *F*(6, 36) = 4.44; *p* = 0.002). There was no effect for gender as a covariable (Wilks’ lambda = 0.746; *F*(6, 36) = 2.04; *p* = 0.085).

Between-subjects effects are shown in Table 4. Self-evaluated insecurity towards technical (*F*(1, 41) = 11.87; *p* = 0.001), legal (*F*(1, 41) = 21.61; *p* ≤ 0.0001), and ethical (*F*(1, 41) = 5.90; *p* = 0.002) aspects was reduced after the training. In addition, the perception of being able to critically reflect on the opportunities and risks of the digital transformation showed a significant effect and was enhanced after the training (*F*(1, 41) = 5.90; *p* = 0.002). Preparedness for the digital transformation did not differ before or after the training (*F*(1, 41) = 3.57; *p* = 0.066) and remained at a low level at the end of the training (*M* = 2.8, *SD* = 1.0).

Standardized mean differences revealed that there were large effects for the insecurity towards technical (*g* = −0.99), legal (*g* = −1.40) and ethical (*g* = −0.82) aspects of the digital transformation [57]. The effect on the critical reflection of opportunities and risks could be estimated as medium-sized (*g* = 0.60). This was in line with the partial eta-square *η_p_^2^* values in the MANCOVA, which conveyed that a proportion of 13% to 35% of the variance was explained by the training condition for these variables (see Table 4).

## 4. Discussion

The present study investigated a blended-learning on-the-job training for practicing physicians in Germany aiming to empower them for the changes due to the digital transformation. The training adopted a structured blended-learning approach, covering basic and critical aspects of the digital transformation in health care. We offered a definition of physicians’ empowerment during this transition, including the enhancement of digital competence. Besides the knowledge domain, we particularly focused on the physicians’ attitudes regarding their perceived empowerment for this process. Our results were generally in line with our hypothesis and indicated that the training was both able to enhance the self-evaluated knowledge base of practicing physicians substantially as well as to shift their attitudes towards a more empowered mindset.

**Hypothesis H1—Knowledge.** The first hypothesis regarding the physicians’ self-evaluated knowledge about central content domains of the digital transformation was completely confirmed. Participants started from a low- to medium-sized level of knowledge, which was enhanced after the training in all nineteen content domains. Knowledge gains differed substantially between domains, with smaller gains for topics such as data protection and data security or basic terms of artificial intelligence and large gains for telematics infrastructure, telemedicine, and digital health solutions. All effects can be classified as large [57], suggesting that the mean level of self-evaluated knowledge after the training was 1.06 to 2.82 standard deviations above the mean level before the training. This is both within and above the reported range of effect sizes for blended-learning trainings in the health care professions [58]. Thus, by means of the presented training, a substantial and meaningful gain of knowledge about digital transformation topics was achieved. For the present study, we deliberately focused on the cognitive part of the Miller pyramid (declarative knowledge “knows” [59]), as practicing physicians currently mostly have a small knowledge base regarding the aspects of the digital transformation [20,21,60,61]. Whether the reported knowledge gain is enough to act confidently and critically in the digitalized health care environment [8] will require further studies. Further, it should be examined if the reported effects are stable and lead to practical benefits for patients and physicians themselves. Moreover, higher-order steps of the Miller pyramid [59] need to be integrated into further trainings.

**Hypothesis H2—Attitudes.** The second hypothesis was partly confirmed by the data. Physicians reported a reduced feeling of insecurity towards technical, legal, and ethical aspects of the digital transformation in health care and a higher sensitivity for its opportunities and risks. However, contrary to our hypothesis, the physicians’ feeling of being prepared for the transformation did not change significantly and remained on a low level after the training. The physicians’ openness to applying innovative technologies did not change during the training either. Interestingly, it was already on an elevated level at the beginning, pointing towards a biased sample of participants, who, in general, are more open to innovations than are other physicians. In contrast to the self-evaluated knowledge domains, the effects for the changes in attitudes during the training were smaller. Still, they differed between 0.60 and 1.4 standard deviations from their starting points, representing medium-to-large effects [57].

**Digital competence.** According to the EU-definition of digital competence, physicians need to be confident about their participation in the digital change processes as well as to be able to critically use technologies [8]. Only during the transformative integration of knowledge, attitudes, and skills can digital competence be developed and have an effect in the physicians’ workplace [62]. Our results show that, at least right after the training, participating physicians reported a mindset that supports this transformative integration [62]. We are aware of the fact that the concept of competence includes procedural skills as well [27]. For a start, we did not include skills in the study. It was known from the literature [20,21,60,61] as well as from the needs assessment in the context of the six-step approach by Kern [15,50] that a majority of physicians currently still have a low level of digital competence. That is why current continuing trainings need to focus on establishing a strong competence foundation, which can be successively built upon with higher-order steps of the Miller pyramid [59]. We take this into account and plan, among other things, to simulate the digitalized health care environment of the German telematics infrastructure in further trainings.

**Reflection on gender.** In the current study, we did not find significant gender effects for the levels of self-evaluated knowledge or attitudes. Nevertheless, descriptive statistics point towards higher reported knowledge levels for men compared to women as well as higher insecurities before training reported by women compared to men. Additional post hoc correlation analysis revealed that there indeed is a significant relationship between attitudes and gender. This would be in line with literature about confidence and self-estimation and gender. Men tend to overestimate competency; women tend to underestimate [63,64]. We suggest that this should be considered in further studies. It is possible that the present sample size was too small to detect gender effects.

**Practical relevance and generalizability.** We presented results that demonstrate how self-evaluated knowledge and attitudes of practicing physicians towards the digital transformation in health care might be enhanced. Additionally, we know that the participants were highly satisfied with the training, and all of them would recommend it to their colleges [15]. Furthermore, the training is going to be provided by the medical association of the federal state of Saxony-Anhalt. The future will show if physicians are willing to take the personal responsibility and participate in the continuing medical education that is offered. As other studies show, a well-tailored, accurate product and application is necessary [60,65]. The strong orientation of our training towards the needs of the target group with a reference to a local and highly work-related context should enhance this fitting accuracy.

Nevertheless, a single on-the-job training is not sufficient to prepare health care professionals for the digital transformation, as is particularly underlined by the low reported preparedness in our study. Diverse variables of the future workplace are still undefined, and the difference between a high-tech future and a currently under-digitized workplace is still high [60]. However, our results show that practicing physicians need to be addressed by education as well as by politics and industry as the important stakeholders and gatekeepers that they are in the health care system [9,10,11,12,13,36]. There is a high demand on catching up in order to educate and train the user of digital technologies not only in university but also in professional contexts. Our own research demonstrated anecdotally that developers and companies currently do not have in mind that physicians need to be trained and engaged in technologies, e.g., in order to prescribe a digital health solution to a patient or to be able to work correctly with electronic records or electronic prescriptions [23]. This is in line, too, with other studies as well as reports of participants of our trainings relating to frustration about the way of implementing digital technologies and about outdated systems in Germany [66]. Consequently, trainings such as the one presented are important, as they also strengthen resilience against (ongoing) disadvantageous working conditions. Moreover, they are only one part in a bigger picture, such as the one suggested by the European commission for Europe [28] or the NHS for the United Kingdom [67]. Next to the question of financing continuing trainings about digital transformation topics, how it may be possible to offer trainings comprehensively for all physicians, in terms of locality and resources at their workplaces, should also be discussed [9]. Empowerment of physicians is necessary. It needs to be stressed that every political strategy towards the health care system in the digital transformation should include the aim of an empowered health care workforce of physicians who are supported to take decisions confidentially, critically, and proactively in the favor of their patients’ and their own health.

**Limitations.** Some additional factors need to be considered regarding generalization of the results. The investigated sample of practicing physicians is not representative for all physicians. Instead, they are those who identified an individual knowledge gap regarding the digital transformation [68] and were motivated to engage in the learning till the completion of the training [68]. This may cause bias and might lead to an overestimation of the effects compared to a general population of practicing physicians. Additionally, the sample with *n* = 32 is only small and should be increased during further studies.

Regarding the measurement of the dependent variables, it must be taken into account that they were all self-reported variables. In addition, the survey after the training took place right at the end, so we do not know how the perception of knowledge develops in the long run. Therefore, additional studies should consider objective tests of knowledge and/or competency as well as include follow-up surveys to assess the sustainability of the knowledge gain and its application in the workplace.

It needs to be stated as well that the individual link of the pre- and post-training sample was not possible, as anonymity during the training would not have been assured otherwise. Therefore, the analysis took place under the assumption of independent samples even though it was actually a matter of repeated measures. Thus, the presented results potentially underestimate the results of dependent samples.

The presented training is specifically tailored according to the needs of the target group of physicians in hospitals in Saxony-Anhalt and contains references to local projects and initiatives as well as to the German digital health infrastructure (telematics infrastructure). A generalizability to other federal states or countries is not self-evident.

## 5. Conclusions

Currently, practicing physicians possess a low level of knowledge about important aspects of the digital transformation and do not feel well-prepared for this fundamental change process. Their self-evaluated knowledge together with their awareness for their personal possibilities to act within a digitalized health care environment were substantially enhanced by means of the presented blended-learning on-the-job training. Further, physicians’ attitudes shifted to a more empowered mindset during the training. Thus, a structured on-the-job training might be a first relevant approach to empower practicing physicians for working in a digitalized health care environment. Nevertheless, additional support needs to be offered. Only empowered physicians can empower patients.

## Figures and Tables

**Figure 1 ijerph-19-12991-f001:**
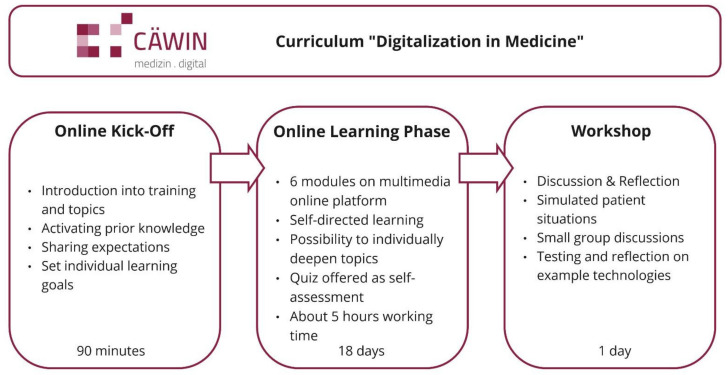
The three parts of the training “Digitalization in Medicine”.

**Figure 2 ijerph-19-12991-f002:**
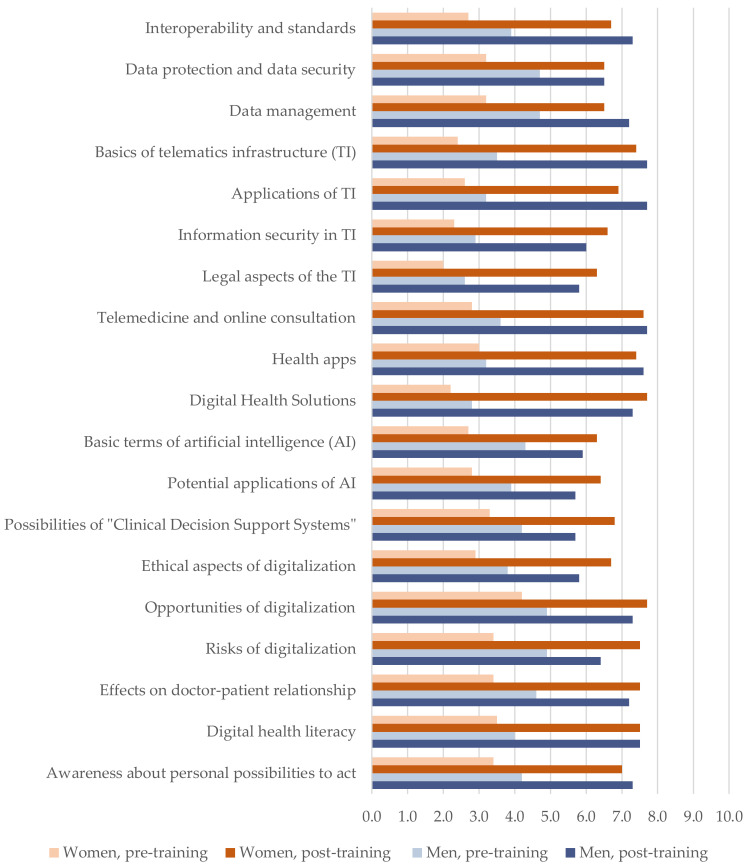
Physicians’ self-evaluated knowledge about central content domains of the digital transformation, pre-training vs. post-training for *n* = 32 participants compared for gender.

**Figure 3 ijerph-19-12991-f003:**
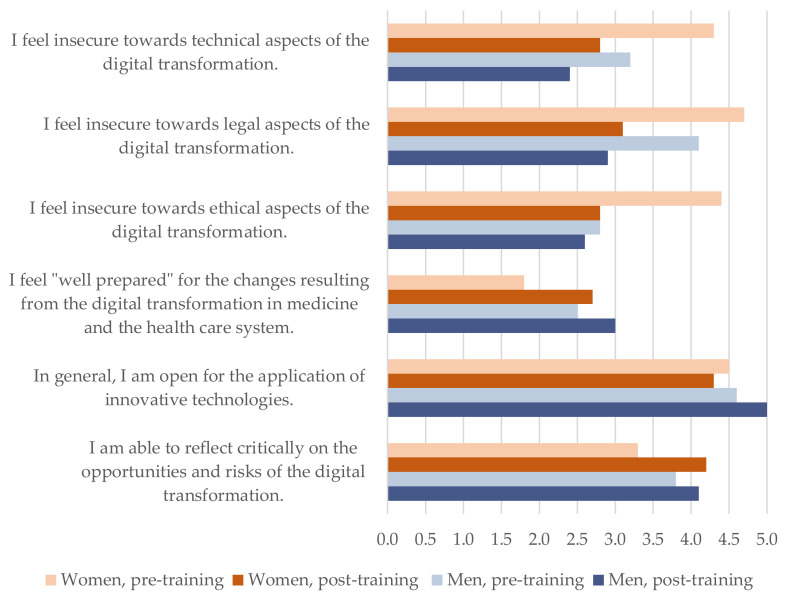
Self-evaluated attitudes regarding physicians’ empowerment for the digital transformation pre-training vs. post-training for *n* = 32 participants compared for gender.

**Table 1 ijerph-19-12991-t001:** Description of the sample across the three training sessions (T1: August 2021, T2: November 2021, T3: March 2022).

	T1	T2	T3	Sum
Number of participants (training completed)	7	11	14	32
Women	2	4	7	13
Men	5	7	7	19
Resident physician	2	4	6	12
Specialist	0	2	4	6
Senior physician	3	3	3	9
Chief physician	1	1	0	2
Professional position not indicated	1	1	1	3

**Table 2 ijerph-19-12991-t002:** Descriptive statistics for self-evaluated knowledge and attitudes towards the digital transformation (sample size *n*, mean *M*, and standard deviation *SD*) as a function of condition (pre-training vs. post-training); standardized mean differences are indicated by Hedges’ *g*.

		Pre-Training			Post-Training			*g*
Nr.	Item	*n*	*M*	*SD*	*n*	*M*	*SD*	
	Knowledge							
**1**	Interoperability and standards	32	3.5	2.3	31	7.2	1.3	1.98
**2**	Data protection and data security	32	4.3	2.3	31	6.5	1.8	1.06
**3**	Data management	32	4.4	2.1	31	6.9	1.7	1.31
**4**	Basics of telematics infrastructure (TI)	32	3.2	1.8	31	7.7	1.7	2.57
**5**	Applications of TI	31	3.0	1.9	31	7.5	2.0	2.31
**6**	Information security in TI	32	2.7	1.5	31	6.5	2.0	2.16
**7**	Legal aspects of the TI	32	2.4	1.2	31	6.2	2.0	2.31
**8**	Telemedicine and online consultation	32	3.3	1.8	31	7.7	1.5	2.65
**9**	Health apps	32	3.3	1.8	31	7.6	1.7	2.46
**10**	Digital health solutions	32	2.8	1.7	31	7.6	1.7	2.82
**11**	Basic terms of artificial intelligence (AI)	32	3.9	2.0	31	6.3	2.1	1.17
**12**	Potential applications of AI	32	3.7	2.0	31	6.2	2.1	1.22
**13**	Possibilities of clinical decision-support systems	31	4.0	2.2	31	6.4	1.8	1.19
**14**	Ethical aspects of digitalization	32	3.5	1.6	31	6.4	1.4	1.93
**15**	Opportunities of digitalization	32	4.7	2.1	31	7.5	1.5	1.53
**16**	Risks of digitalization	32	4.4	1.9	31	7.0	1.5	1.52
**17**	Effects on doctor–patient relationship	32	4.1	2.0	31	7.4	1.4	1.91
**18**	Digital health literacy	32	3.8	1.9	31	7.6	1.6	2.16
**19**	Awareness about personal possibilities to act	32	3.9	2.1	31	7.3	1.5	1.86
	**Attitude**							
**1**	I feel insecure towards technical aspects of the digital transformation.	24	3.7	1.0	25	2.6	1.2	−0.99
**2**	I feel insecure towards legal aspects of the digital transformation.	23	4.3	0.7	25	3.0	1.1	−1.40
**3**	I feel insecure towards ethical aspects of the digital transformation.	24	3.5	1.2	25	2.6	1.0	−0.82
**4**	I feel well-prepared for the changes resulting from the digital transformation in medicine and the health care system.	24	2.3	1.2	25	2.8	1.0	0.45
**5**	In general, I am open for the application of innovative technologies.	24	4.6	0.6	25	4.6	0.8	0.00
**6**	I am able to reflect critically on the opportunities and risks of the digital transformation.	24	3.7	0.8	25	4.1	0.5	0.60

**Table 3 ijerph-19-12991-t003:** Results of the between-subjects analysis of the MANOVA for self-evaluated knowledge domains as dependent variable.

Dependent Variable	*df*	*F*	*p*	*η_p_* ^2^
Interoperability and standards	1, 55	71.29	<0.0001	0.56
Data protection and data security	1, 55	22.16	<0.0001	0.29
Data management	1, 55	35.25	<0.0001	0.39
Basics of telematics infrastructure (TI)	1, 55	97.09	<0.0001	0.64
Applications of TI	1, 55	75.14	<0.0001	0.58
Information security in TI	1, 55	62.79	<0.0001	0.53
Legal aspects of the TI	1, 55	80.62	<0.0001	0.59
Telemedicine and online consultation	1, 55	104.61	<0.0001	0.66
Health apps	1, 55	102.49	<0.0001	0.65
Digital health solutions	1, 55	122.54	<0.0001	0.69
Basic terms of artificial intelligence (AI)	1, 55	22.16	<0.0001	0.29
Potential applications of AI	1, 55	26.96	<0.0001	0.33
Possibilities of clinical decision-support systems	1, 55	19.76	<0.0001	0.26
Ethical aspects of digitalization	1, 55	53.84	<0.0001	0.49
Opportunities of digitalization	1, 55	32.10	<0.0001	0.37
Risks of digitalization	1, 55	31.37	<0.0001	0.36
Effects on doctor–patient relationship	1, 55	46.08	<0.0001	0.46
Digital health literacy	1, 55	63.57	<0.0001	0.54
Awareness about personal possibilities to act	1, 55	46.96	<0.0001	0.46

**Table 4 ijerph-19-12991-t004:** Results of the between-subjects analysis of the MANOVA for attitudes towards the digital transformation as dependent variable.

Dependent Variable	*df*	*F*	*p*	*η_p_* ^2^
Insecurity towards technical aspects	1, 41	11.87	0.001	0.22
Insecurity towards legal aspects	1, 41	21.61	<0.0001	0.35
Insecurity towards ethical aspects	1, 41	5.90	0.020	0.13
Preparedness for digital transformation	1, 41	3.57	0.066	0.08
Open for innovative technology	1, 41	0.18	0.675	0.00
Critical reflection on opportunities and risks	1, 41	6.21	0.017	0.13

## Data Availability

Data can be obtained from the corresponding author upon reasonable request.

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
