# Peer review of "Empowerment for the Digital Transformation: Results of a Structured Blended-Learning On-the-Job Training for Practicing Physicians in Germany"

_ijerph, 2022, doi:10.3390/ijerph192012991_

Round 1

Reviewer 1 Report

Dear Editors, dear Authors, 

The article “Empowerment for the digital transformation: Results of a structured blended-learning on-the-job training for practicing physicians in Germany” addresses a highly relevant aspect of the current digital transformation of medicine. While the article the article overall displays a good quality, some changes and additions should be made prior to a decision concerning publication: 

·       The nature of the investigation should be consistently named in the article. Self-evaluation has its limitations, and it should be clear to the readers at all stages of the article that this is the used method. Namely it should be addressed in formulation of the hypothesis, during the results presentation (text and figures) and discussion. 

·       The article would clearly benefit from a more detailed description of the curriculum, its learning objectives and the competencies adressed, the didactic approaches and the schedule. This could be added in text form and/or table/figure. 

·       It should be referenced in which part the presented qualification is in accordance with the published curriculum by the German Medical Association and in which part it does include other/alternative parts. 

·       Based on the curriculum of the German Medical Association, a competency-based, blended learning CME course "Competent for Medicine in the Digital Age (in German: Kompetent für die Medizin im digitalen Zeitalter)" is offered since January 2020. This should be included in the introduction when disussing existing formats. The modules are digital communication, mHealth and smart devices, telemedicine and telematics, health data as well as artificial intelligence and clinical decision support. Participants should be "able to effectively use knowledge, skills, and attitudes related to digital medicine after the training." The training includes approaches to promote critical digital health literacy / digital competencies. For example, participants should be enabled to critically evaluate the benefits and risks of medical apps and smart devices and to use them in a patient-oriented manner (Kompetent für die Medizin im digitalen Zeitalter. Ärztekammer Berlin. Berlin.)

·       The references concerning the undergraduate curriculum for medical students (reference 22, 37) only describe the individual modules. The complete curriculum and the evaluation are described in the following publication:

o   Digital skills for medical students – qualitative evaluation of the curriculum 4.0 “Medicine in the digital age”. GMS J Med Educ. 2020;37(6):Doc60. DOI: 10.3205/zma001353, URN: urn:nbn:de:0183-zma0013535, 

Overall, I would advise for a (minor) revision and resubmission of the article. 

Author Response

Dear Reviewer 1,

Thank you for reviewing our article and for your constructive comments and suggestions. We addressed the points you mentioned in the following way:

It was suggested to make it consistently explicit that the investigation is based on self-evaluation of the participants. Therefore, we specified the description of knowledge and attitude in all parts of the paper as "self-evaluated knowledge/ attitudes/ indicators".

We provide a more detailed description of the presented curriculum, by means of a figure (see page 4), as well as longer description of the learing goals:

  • "At the end of the training, participants should have a basic understanding of central concepts and developments of the digital transformation and their interrelation, as well as their impact on their own work. They should be enabled to reflect critically on chances and risks of the digitalization during their daily medical work and to advise their patients solidly in favor of their needs. Finally, the training aims to raise awareness to develop a personal position towards the digital transformation and to actively take part in this process."

The modules titles are given and some details were added:

  • "They flexibly completed the following six modules on a modern multimedia online platform: 1) Important principles and definitions; 2) Telematics infrastructure; 3) Digital tools; 4) Artificial intelligence and big data; 5) Ethics in digitalization; 6) Physicians and patients. The modules  contained structured micro learning units [52], conveying practical knowledge relevant in the daily work environment of the physicians, mainly by means of video interviews with local experts and practitioners. Complementary, the online platform contained suggestions to individually deepen the knowledge on certain topics. Also, a knowledge quiz was offered as formative self-assessment of the acquired knowledge. The online learning phase was passed parallel to the participants’ job in hospital."

We included a short comparision of the presented curriculum with the curriculum of the German Medical Association:

  • "Besides the general and targeted needs assessment, the presented curriculum is based on the above mentioned basic curriculum by the German Medical Association [34]. Mainly, the topics of both curricula are comparable, and both are certified for 24 CME points. Whereas the German Medical Association integrates its topics into a basic and an advanced module, starting with the telematics infrastructure, the content of the presented curriculum is grouped according to six modules. The first module conveys basic IT-related principles (e.g., interoperability, data security, databases), as premises for comprehension of the other topics. The curriculum in Saxony-Anhalt additionally considers local expertise, such as clinical decision making and use cases developed as part of the Medical Informatics Initiative, funded by the German Federal Ministry of Education and Research."

We included the two papers that were suggested, into the introduction:

Nr. 24: Kuhn, S.; Müller, N.; Kirchgässner, E.; Ulzheimer, L.; Deutsch, K.L. Digital skills for medical students - qualitative evaluation of the curriculum 4.0 "Medicine in the digital age". GMS Journal for Medical Education 2020, 37, Doc60

  • "The need for structured qualifications in digital transformation competencies has internationally been acknowledged in the medical education domain [1–4,9,14,22–24]."
  • "Internationally, many programs and efforts invest in the implementation of digital competencies and data literacy into the undergraduate medical curricula [9,24,31,32]."

Nr. 33: Kuhn, S. Kompetent für die Medizin im digitalen Zeitalter. Berliner Ärzt:innen 2020, 15–18.

  • "In Germany, there are few examples [e.g. 33] that implement a basic curriculum proposed by the German Medical Association [34]."

Thank you again for your valuable comments.

Reviewer 2 Report

This is a well written manuscript which provides interesting information about structured training as a potential factor that may enhance knowledge and attitudes of practicing physicians towards the digital transformation. These findings are worthy of publication in IJERPH but require minor revisions.

1. Lines 171-172. Did the authors consider including age as a covariate in the MANCOVA?

2. Line 177. March 2021?

3. Line 187. Please explain specifically why “position was not included as covariable owing to small group sizes”?

4. Table 2. I suggest providing brief descriptions of these symbols (“N” an “M”) in the paragraph or in the table caption.

5. Line 259. The p values (legal and ethical aspects) are not consistent to those presented in Table 4.

6. Typo: Line 276, should be “on-the-job”; Line 300, should be “Whether”.

Author Response

Dear Reviewer 2,

Thank you for reviewing our article and for your constructive comments and suggestions. We addressed the points you mentioned in the following way:

Lines 171-172. Did the authors consider including age as a covariate in the MANCOVA?

  • we did not survey age as variable in the questionnaire. Age is highly correlated with position

Line 177. March 2021?

  • it was changed into "March 2022", when the training actually took place

Line 187. Please explain specifically why “position was not included as covariable owing to small group sizes”?

  • crossing position and gender, the group size would be only n = 2 for some samples, which does not provide enough information for a profound analysis. that is why, we set position as a covariable aside.

Table 2. I suggest providing brief descriptions of these symbols (“N” an “M”) in the paragraph or in the table caption.

  • a brief description was included in the table caption as suggested

Line 259. The p values (legal and ethical aspects) are not consistent to those presented in Table 4.

  • this was corrected in the text

Typo: Line 276, should be “on-the-job”; Line 300, should be “Whether”.

  • this was corrected in the text

Thank you again for your valuable comments on our paper.

Reviewer 3 Report

Dear authors, 

I agree that digital transformation is an emerging healthcare topic and therefore articles around this should be welcome. This study appear well-designed and well-argued although the sample considered is small. I would suggest minor grammar and English language editing and, as there are some spelling errors, I suggest a careful and meticulous revision (i.e. lines 250, 276). 

Author Response

Dear Reviewer 3,

thank you very much for reviewing our manuscript and your helpful comments. We revised the whole manuscript carefully and corrected errors of orthography and grammar.
